# Photocatalytic three-component asymmetric sulfonylation via direct C(sp³)-H functionalization

Shi Cao[1], Wei Hong[1], Ziqi Ye[1] & Lei Gong [1✉]

The direct and selective C(sp³)-H functionalization of cycloalkanes and alkanes is a highly useful process in organic synthesis owing to the low-cost starting materials, the high step and atom economy. Its application to asymmetric catalysis, however, has been scarcely explored. Herein, we disclose our effort toward this goal by incorporation of dual asymmetric photo-catalysis by a chiral nickel catalyst and a commercially available organophotocatalyst with a radical relay strategy through sulfur dioxide insertion. Such design leads to the development of three-component asymmetric sulfonylation involving direct functionalization of cycloalkanes, alkanes, toluene derivatives or ethers. The photochemical reaction of a C(sp³)-H precursor, a SO₂ surrogate and a common α,β-unsaturated carbonyl compound proceeds smoothly under mild conditions, delivering a wide range of biologically interesting α-C chiral sulfones with high regio- and enantioselectivity (>50 examples, up to >50:1 rr and 95% ee). This method is applicable to late-stage functionalization of bioactive molecules, and provides an appealing access to enantioenriched compounds starting from the abundant hydrocarbon compounds.

---

[1] Key Laboratory of Chemical Biology of Fujian Province, iChEM, College of Chemistry and Chemical Engineering, Xiamen University, Xiamen, Fujian, China.
✉email: gongl@xmu.edu.cn

Chiral sulfones are building blocks found in many biologically active compounds and drugs[1], and sulfones bearing stereocenters at the α-position or β-position have shown great potential in pharmaceutical and clinical applications (Fig. 1a)[2–4]. For example, remikiren is a renin inhibitor used for the treatment of hypertension[5], apremilast has been approved by Food and Drug Administration (FDA) as an oral drug to treat active psoriatic arthritis and plaque psoriasis[6]. Chiral sulfones play important roles in organic synthesis as auxiliaries, ligands, and synthetic intermediates[7,8]. Great efforts have been devoted to the development of the practical synthesis of enantioenriched sulfones[9]. However, there are few asymmetric catalytic approaches to chiral sulfones with α-carbon stereocenters, and those typically rely on noble-metal-catalyzed asymmetric hydrogenation. Neighboring sulfonyl groups have the potential to stimulate racemization of the non-quaternary stereocenters, thereby synthetic approaches to such compounds which can be performed under mild conditions are strongly desired[10–14].

The direct C(sp³)-H functionalization of cycloalkanes and alkanes is a highly useful process due to the low-cost starting materials, the high step, and atom economy (Fig. 1b)[15,16]. Methods to control site-selectivity towards C-H bonds at the specific positions of unactivated cycloalkanes or alkanes (without the assistance of any directing group)[17–30], at the same time combine with asymmetric catalysis, provide appealing opportunities for the construction of high value-added chiral molecules, but remains a remarkable challenge[31]. In this context, rhodium-carbene-induced C–H insertion has emerged as an elegant strategy for alkylation of primary, secondary, ternary alkanes, and cycloalkanes, in which chiral dirhodium catalysts with well-designed and tailored ligands are employed for both site recognition and asymmetric induction[32–37].

Recently, we found a photocatalytic enantioselective alkylation reaction of *N*-sulfonylimines with benzylic or alkyl C(sp³)-H precursors enabled by copper-based asymmetric catalysis and organophotocatalysis. This radical-mediated method however relies on very specific reaction partners and is extremely ineffective for conversions of cycloalkanes[38]. Practical methods which can be applied to common substrates are still undeveloped. To achieve this goal, suppression of the side reactions such as self-coupling or elimination of radical intermediates and powerful stereocontrol in the radical process is required. Sulfur dioxide and its surrogates have been recognized as useful acceptors for alkyl or aryl radicals in many thermal and photochemical reactions[39–46]. The resulting sulfonyl radicals have a longer life time which might be beneficial for asymmetric induction[47]. Hence, we questioned whether incorporation of sulfur dioxide insertion with appropriate dual asymmetric photocatalysis[48–51] would allow us to find an effective approach for stereoselective transformations starting from most abundant hydrocarbon compounds. On the basis of these considerations, a photocatalytic asymmetric three-component sulfonylation reaction of a cycloalkane (an alkane, a toluene derivative, or an ether), an SO₂ surrogate, and a common Michael acceptor was developed, whose stereochemistry was governed by a nickel catalyst of a well-tailored chiral bisoxazoline ligand. This method provides straightforward and economic access to biologically interesting α-C chiral sulfones (>50 examples, up to 50:1 rr and 95% ee, Fig. 1c).

## Results

**Reaction development**. α,β-Unsaturated carbonyl compounds are one class of common Michael acceptors in organic synthesis. Initially, we chose cyclohexane (**1a**) as the model substrate, α,β-unsaturated carbonyl compound bearing an *N*-acylpyrazole moiety (**2a**) as the reaction partner, 5,7,12,14-pentacenetetrone

(**PC1**) as the hydrogen atom transfer (HAT) photocatalyst[38,52], and the bisoxazoline nickel complex generated in situ ([**L1**-**Ni**]) as the chiral catalyst (Table 1). The reaction of **1a** and **2a** failed to produce an additional product under visible light conditions (entry 1). We next tested several SO₂ surrogates in the photochemical system and found that 1,4-Diazabicyclo[2.2.2]octane-1,4-diium-1,4-disulfinate (DABCO·(SO₂)₂) was the appropriate reaction component (entries 2,3). Under irradiation with a 24 W blue LEDs lamp ($\lambda_{max}$ = 455 nm) at 20°C under argon, the reaction of **1a**, **2a**, and DABCO·(SO₂)₂ in dichloroethane delivered the chiral sulfone product (**3a**) in 58% conversion and with only 3% ee (entry 3). α,β-Unsaturated carbonyl substrates bearing a different auxiliary group (Z) were examined, and it was found that *N*-acylpyrazole (**2a**) was a suitable substrate (entries 4–7). Other photocatalysts (**PC2**–**PC4**) failed to catalyze this transformation (entries 8–10). In order to improve the reaction efficiency and enantioselectivity, a range of chiral ligands (**L2**–**L11**) were screened (entries 11–20). Chiral bis(oxazoline) ligands (**L2**–**L4**) and a tridentate ligand (**L5**), which have been successfully used for asymmetric induction in some radical-mediated transformations[53–56], were found to be ineffective in this photocatalytic reaction (entries 11–14), but replacement of **L1** with an indane-derived BOX ligand (**L6**) led to full conversion and significantly increased enantiomeric excess (82% ee) (entry 15). Based on this observation, we modified this type of ligands and synthesized several sterically more bulky analogs for the creation of a more precise chiral environment (**L7**–**L9**). One of these, **L7** was identified as the best ligand with regard to the yield and enantioselectivity (86% ee) (entry 16). Finally, the reaction was further improved by reducing the reaction temperature and increasing the loading of chiral catalyst, which provided **3a** with a full conversion and 95% ee (entry 23). The reaction with only 1.0 equiv. of cyclohexane also proceeded smoothly at a lower but reasonable reaction rate (52% conversion within 96 h) and similar enantioselectivity of 93% ee (entry 24). Such conditions would be useful with more expensive C(sp³)-H precursors such as natural products or drug molecules.

**Reaction scope investigation**. With the optimal reaction conditions in hand, we investigated the substrate scope of this photocatalytic asymmetric three-component sulfonylation reaction (Fig. 2). It was revealed that unsubstituted cycloalkanes such as cyclohexane (**1a**), cyclopentane (**1b**), cycloheptane (**1c**), cyclooctane (**1d**), and cyclododecane (**1e**) were excellent substrates, delivering the chiral sulfone products (**3a**–**3e**) in 62–74% yield and with 91–95% ee. A trisubstituted cycloalkane, 1,3,5-trimethylcyclohexane (**1f**) exhibited high regioselectivity towards the tertiary C(sp³)-H bonds (>50:1 rr) and provided lower enantioselectivity (78% ee). The observation of high regioselectivity was consistent with an inherent difference of bond dissociation energies (BDEs) of secondary and tertiary C(sp³)-H bonds, as well as stability of the corresponding carbon radicals. The crystal structure of product **3b** revealed the absolute configuration (*R*) of the major enantiomer, and the absolute configuration of the other products was assigned accordingly. Adamantane has a significantly higher 3° C(sp³)-H bond dissociation energy of 99 kcal mol⁻¹ caused by its rigid cage structure, which exceeds the bond dissociation energy of its 2° C(sp³)-H bonds (96 kcal mol⁻¹) and those of most other hydrocarbons[57]. The unusually strong tertiary C(sp³)-H bonds of adamantine-type compounds present a remarkable challenge to their selective tertiary C-H functionalization[58], but such transformations proceeded very well in our photochemical system. Adamantane (**1g**), 1-methyladamantane (**1h**), 1-ethyladamantane (**1i**), and 1,3-dimethyladamantane (**1j**) were all selectively functionalized at the

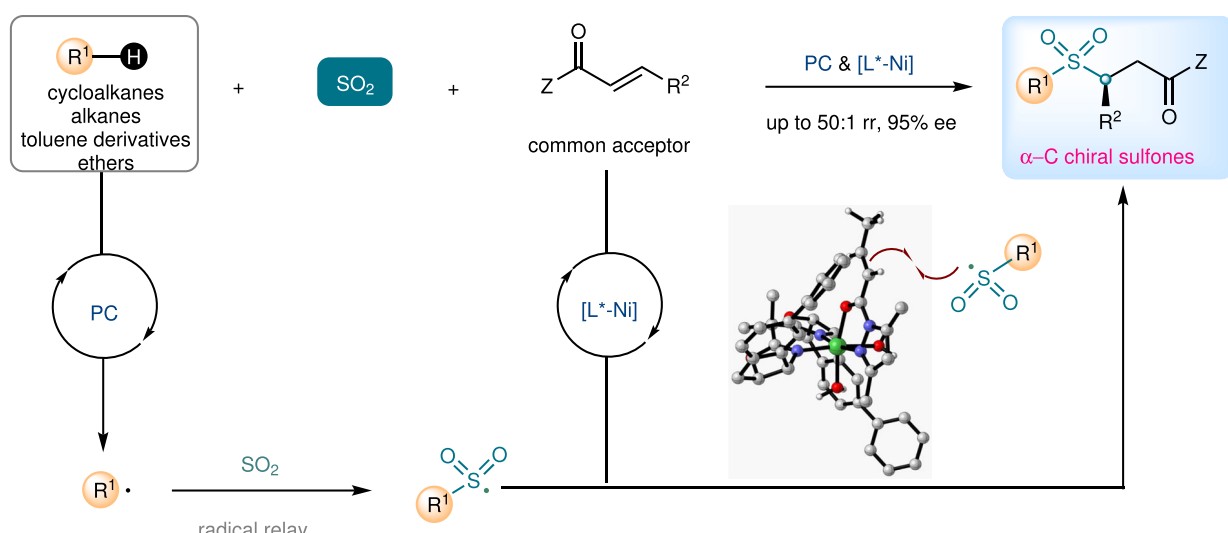

**Fig. 1 Overview of this work. a** Bioactive molecules or natural products containing α-C chiral sulfone units. **b** Challenges of direct C(sp³)-H functionalization of cycloalkanes and alkanes. **c** This work: photocatalytic three-component sulfonylation via direct C(sp³)-H functionalization of cycloalkanes and alkanes. BDEs, bond dissociation energies; rr, regioisomeric ratio; ee, enantiomeric excess; PC, photocatalyst; Z, auxiliary group.

tertiary C(sp³)-H position, affording the products (**3g–3j**) as single regioisomers with 84–90% ee. These results indicate that the catalyst-control also plays a crucial role in the selectivity in addition to the inherent difference of bond dissociation energies and carbon radical stabilities. The reaction exhibited high functional group compatibility, which would be desirable for its synthetic application to structurally diverse compounds. For example, adamantine derivatives bearing a fluoro- (**1k**), chloro- (**1l**), bromo- (**1m**), cyano- (**1n**), ester- (**1o**), keto- (**1p**), or

hydroxyl- (**1q**) substituent all provided a reasonable yield (59–71%), perfect regioselectivity (>50:1), and satisfactory enantioselectivity (82–91%). Moreover, the photochemical reactions of tertiary alkanes also proceeded smoothly under the standard conditions and delivered the desired chiral sulfones (**3r–3u**) as exclusive regioisomers with 75–82% ee.

To further investigate the generality of this method, other C(sp³)-H precursors were examined. The reaction with toluene and its derivatives was more rapid than those of cycloalkanes and

**Table 1 Optimization of reaction conditions[a].**

| Entry | Metal salt | Ligand | PC | SO$_2$ surrogates | Substrate | T (°C) | Product | Conv. (%)[b] | Ee (%)[c] |
|---|---|---|---|---|---|---|---|---|---|
| 1 | Ni(ClO$_4$)$_2$·6H$_2$O | L1 | PC1 | none | 2a | 20 | – | 0 | n.a. |
| 2 | Ni(ClO$_4$)$_2$·6H$_2$O | L1 | PC1 | Na$_2$S$_2$O$_5$ | 2a | 20 | 3a | 0 | n.a. |
| 3 | Ni(ClO$_4$)$_2$·6H$_2$O | L1 | PC1 | DABCO•(SO$_2$)$_2$ | 2a | 20 | 3a | 58 | 3 |
| 4 | Ni(ClO$_4$)$_2$·6H$_2$O | L1 | PC1 | DABCO•(SO$_2$)$_2$ | 2b | 20 | 3b | 0 | n.a. |
| 5 | Ni(ClO$_4$)$_2$·6H$_2$O | L1 | PC1 | DABCO•(SO$_2$)$_2$ | 2c | 20 | 3c | 0 | n.a. |
| 6 | Ni(ClO$_4$)$_2$·6H$_2$O | L1 | PC1 | DABCO•(SO$_2$)$_2$ | 2d | 20 | 3d | 0 | n.a. |
| 7 | Ni(ClO$_4$)$_2$·6H$_2$O | L1 | PC1 | DABCO•(SO$_2$)$_2$ | 2e | 20 | 3e | 0 | n.a. |
| 8 | Ni(ClO$_4$)$_2$·6H$_2$O | L1 | PC2 | DABCO•(SO$_2$)$_2$ | 2a | 20 | 3a | 0 | n.a. |
| 9 | Ni(ClO$_4$)$_2$·6H$_2$O | L1 | PC3 | DABCO•(SO$_2$)$_2$ | 2a | 20 | 3a | 0 | n.a. |
| 10 | Ni(ClO$_4$)$_2$·6H$_2$O | L1 | PC4 | DABCO•(SO$_2$)$_2$ | 2a | 20 | 3a | 0 | n.a. |
| 11 | Ni(ClO$_4$)$_2$·6H$_2$O | L2 | PC1 | DABCO•(SO$_2$)$_2$ | 2a | 20 | 3a | 47 | 13 |
| 12 | Ni(ClO$_4$)$_2$·6H$_2$O | L3 | PC1 | DABCO•(SO$_2$)$_2$ | 2a | 20 | 3a | 53 | 11 |
| 13 | Ni(ClO$_4$)$_2$·6H$_2$O | L4 | PC1 | DABCO•(SO$_2$)$_2$ | 2a | 20 | 3a | 49 | 21 |
| 14 | Ni(ClO$_4$)$_2$·6H$_2$O | L5 | PC1 | DABCO•(SO$_2$)$_2$ | 2a | 20 | 3a | 37 | 11 |
| 15 | Ni(ClO$_4$)$_2$·6H$_2$O | L6 | PC1 | DABCO•(SO$_2$)$_2$ | 2a | 20 | 3a | quant. | 82 |
| 16 | Ni(ClO$_4$)$_2$·6H$_2$O | L7 | PC1 | DABCO•(SO$_2$)$_2$ | 2a | 20 | 3a | quant. | 86 |
| 17 | Ni(ClO$_4$)$_2$·6H$_2$O | L8 | PC1 | DABCO•(SO$_2$)$_2$ | 2a | 20 | 3a | 90 | 84 |
| 18 | Ni(ClO$_4$)$_2$·6H$_2$O | L9 | PC1 | DABCO•(SO$_2$)$_2$ | 2a | 20 | 3a | 85 | 86 |
| 19 | Ni(ClO$_4$)$_2$·6H$_2$O | L10 | PC1 | DABCO•(SO$_2$)$_2$ | 2a | 20 | 3a | quant. | 82 |
| 20 | Ni(ClO$_4$)$_2$·6H$_2$O | L11 | PC1 | DABCO•(SO$_2$)$_2$ | 2a | 20 | 3a | quant. | 79 |
| 21 | Ni(ClO$_4$)$_2$·6H$_2$O | L7 | PC1 | DABCO•(SO$_2$)$_2$ | 2a | 0 | 3a | 77 | 95 |
| 22 | Ni(ClO$_4$)$_2$·6H$_2$O | L7 | PC1 | DABCO•(SO$_2$)$_2$ | 2a | −20 | 3a | 47 | 95 |
| 23[d] | Ni(ClO$_4$)$_2$·6H$_2$O | L7 | PC1 | DABCO•(SO$_2$)$_2$ | 2a | 0 | 3a | quant. | 95 |
| 24[e] | Ni(ClO$_4$)$_2$·6H$_2$O | L7 | PC1 | DABCO•(SO$_2$)$_2$ | 2a | 0 | 3a | 52 | 93 |

[a]Reaction conditions: **1a** (1.0 mmol, 10 equiv.), **2a–2e** (0.10 mmol, 1.0 equiv.), SO$_2$ surrogate (0.075 mmol, 1.5 equiv.), metal salt (0.010 mmol, 10 mol%), ligand (0.012 mmol, 12 mol%), **PC1–PC4** (0.0050 mmol, 5 mol%), ClCH$_2$CH$_2$Cl (4.0 mL), indicated temperature, 24 W blue LEDs lamp ($\lambda_{max}$ = 455 nm), under argon, see more details for the screening of nickel salts and solvents in Supplementary Table 1.
[b]Conversion determined by $^1$H-NMR.
[c]Ee value determined by chiral HPLC.
[d]Nickel salt (0.015 mmol, 15 mol%), chiral ligand (0.018 mmol, 18 mol%).
[e]Reaction performed with 1.0 equiv. of cyclohexane within 96 h.
*PC* photocatalyst, *Z* auxiliary group, *conv.* conversion, *quant.* quantitative conversion, *n.a.*, not applicable, *DABCO·(SO$_2$)$_2$* 1,4-Diazabicyclo[2.2.2]octane-1,4-diium-1,4-disulfinate.

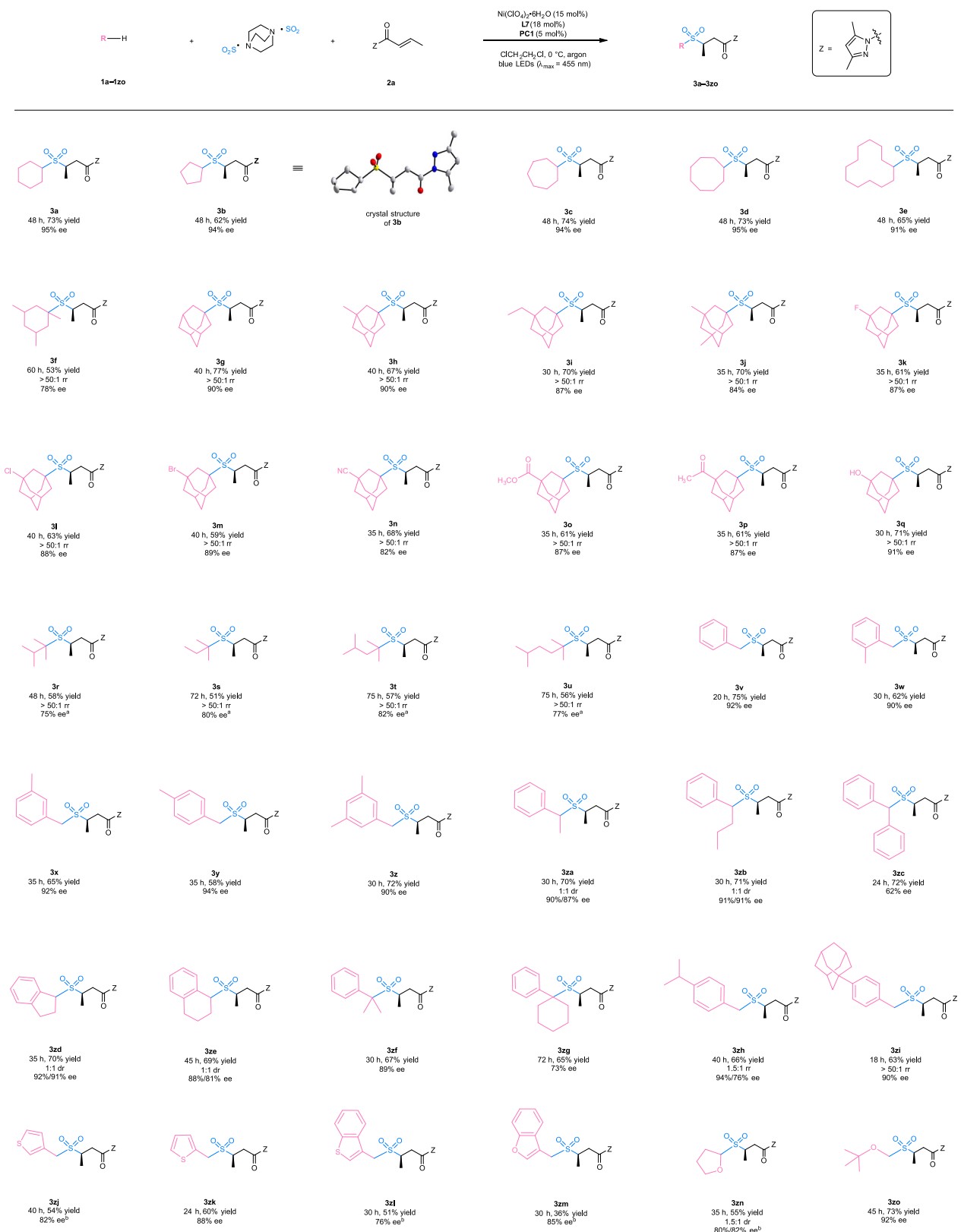

**Fig. 2 Reaction scope of C(sp³)-H precursors including cycloalkanes, alkanes, toluene derivatives, and ethers.** [a]Reaction performed in the presence of alkanes (20 equiv.). [b]Reaction performed at 20 °C. PC, photocatalyst; Z, auxiliary group.

**Fig. 3 Reaction scope of α,β-unsaturated *N*-acylpyrazoles.** [a]Reaction performed at a lower concentration and with a modified substrate ratio, see more details in Supplementary Fig. 1. PC, photocatalyst; Z, auxiliary group.

alkanes. The primary (**1v–1z**), secondary (**1za–1ze**), and tertiary benzylic hydrocarbons (**1zf–1zh**) all worked well and delivered the chiral products (**3v–3zh**) in 58–75% yield and with 62–94% ee. Sterically more demanding substrates such as diphenylmethane (**1zc**) and cyclohexylbenzene (**1zg**) tended to give lower enantioselectivity. The reactions of toluene derivatives containing completing reactive sites displayed some degree of regioselectivity. For example, *p*-isopropyl toluene (**1zh**) provided very moderate regioselectivity of 1.5:1, while 1-(*p*-tolyl)adamantine bearing benzylic C(sp³)-H bonds, secondary and tertiary C(sp³)-H bonds in the admantyl moiety (**1zi**) afforded a single regioisomer with the benzylic functionalization. It was found that heteroaromatic α-C(sp³)-H precursors such as 3-methylthiophene (**1zj**), 2-methylthiophene (**1zk**), 3-methylbenzothiophene (**1zl**), and 3-methylbenzofuran (**1zm**) were compatible with the reaction and gave products with 76–88% ee. Interestingly, ethers such as tetrahydrofuran (**1zn**) and methyl *tert*-butyl ether (**1zo**), which have strong α-C(sp³)-H bonds (BDEs = ~92 kcal mol⁻¹)[59], were also identified as excellent substrates by the yields (55–73%) and enantioselectivities (80–92% ee).

We next evaluated the scope of α,β-unsaturated *N*-acylpyrazoles. As summarized in Fig. 3, β-substituents including a linear alkyl group (products **3zp–3zr**), a branched alkyl group (products **3zs, 3zt**), a cylcoalkyl group (product **3zu**), and an aryl substituent (products **3zv–3zy**) were all compatible with regards to yields (53–71%) and enantioselectivity (64–93% ee). Typically, β-alkyl-substituted substrates gave better enantioselectivity in comparison to those containing a β-aryl group, perhaps due to the trend of Z/E isomerization of the latter under photochemical conditions[60]. For example, in the reaction of **1a** + DABCO · (SO₂)₂ + **2l** → **3zv**, 28% of Z-isomer of **2l** was isolated (see more details in Supplementary Fig. 5). Such byproducts were not observed in the conversions of β-alkyl α,β-unsaturated *N*-acylpyrazoles.

**Mechanistic studies.** Several control experiments were conducted to gain insight into the reaction mechanism (Fig. 4). For example, the addition of a radical quencher (2,2,6,6-tetramethylpiperidine-1-oxyl, TEMPO, 3 equiv.) to the photochemical reaction **1v** +

DABCO · (SO₂)₂ + **2a** → **3v** was found to completely inhibit the transformation to **3v**, instead of affording a TEMPO-carbon radical cross-coupling product (**4**) detected by HRMS analysis. The three-component sulfonylation reaction of 4-(cyclopropylmethyl)-1,1'-biphenyl (**1zp**) under the standard conditions provided a ring-opened product (**5**) in 63% yield and with 95% ee, which further confirmed the reaction pathway via benzylic radicals. Employing cyclopropyl-substituted *N*-acylpyrazole (**2p**) as a substrate for the radical clock experiment provided product **6** in 65% yield and did not give any ring-opened product, thus excluding mechanisms involving the β-carbon radicals of *N*-acylpyrazole complexes[61]. The reaction of **2a** with sodium cyclopentanesulfinate was examined under the standard conditions and failed to produce any desired product **3b** (Fig. 4b). This result indicated that the reaction might not proceed through sulfonyl anion intermediates. Moreover, removal of the nickel catalyst in the reaction of **1v** + DABCO · (SO₂)₂ + **2a** → **3v** led to the failure of product formation, while replacing the nickel catalyst with other Lewis acids such as a copper, zinc, iron, or cobalt complex of the same chiral ligand (**L7**) still afforded product **3v** in a moderate yield (17 − 31%). The cobalt complex with a similar octahedral configuration even gave good enantioselectivity of 80% (Fig. 4c). These results suggested that the nickel complex most likely only provided Lewis acid activation for α,β-unsaturated *N*-acylpyrazoles. Finally, luminescence quenching experiments revealed that the C(sp³)-H precursors such as toluene (**1v**) were capable of quenching the excited state of **PC1** and initiating the radical process (Fig. 4d).

**Mechanistic proposal.** On the basis of the initial experiments and mechanistic studies, we propose a plausible reaction mechanism (Fig. 5a). The chiral nickel catalyst ([**L\*-Ni**]) undergoes fast ligand exchange with the α,β-unsaturated *N*-acylpyrazole (**2**) to generate an intermediate complex (**A**). On the other hand, the organophotocatalyst (**PC**) is excited to its triplet state (**B**), which performs hydrogen atom abstraction from the C(sp³)-H precursor (**1**) to give the semiquinone-type radical intermediate (**C**) and the transient carbon radical (**D**)[52]. The radical (**D**) is rapidly trapped by the sulfur dioxide released in situ to produce the

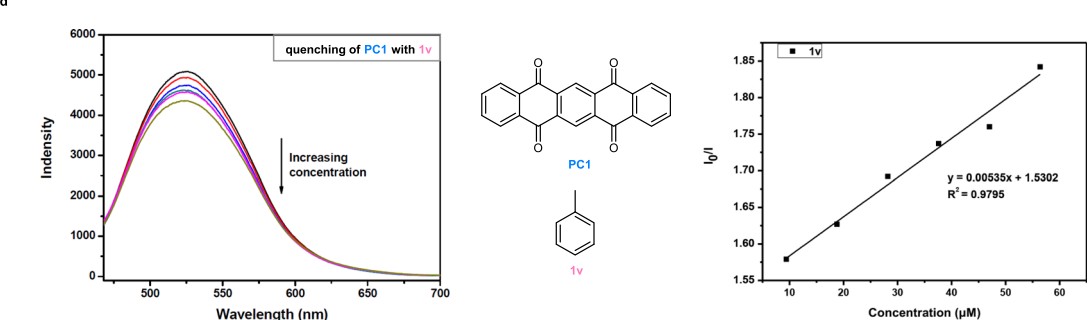

**Fig. 4 Control experiments. a** Probing of carbon radical pathways. **b** Probing of sulfonyl anion pathways. **c** Reactions by other Lewis acids. **d** Luminescence quenching experiments. DABCO • $(SO_2)_2$, 1,4-Diazabicyclo[2.2.2]octane-1,4-diium-1,4-disulfinate; TEMPO, 2,2,6,6-tetramethylpiperidine-1-oxyl; PC, photocatalyst; equiv., equivalent; conv., conversion; n.a., not applicable.

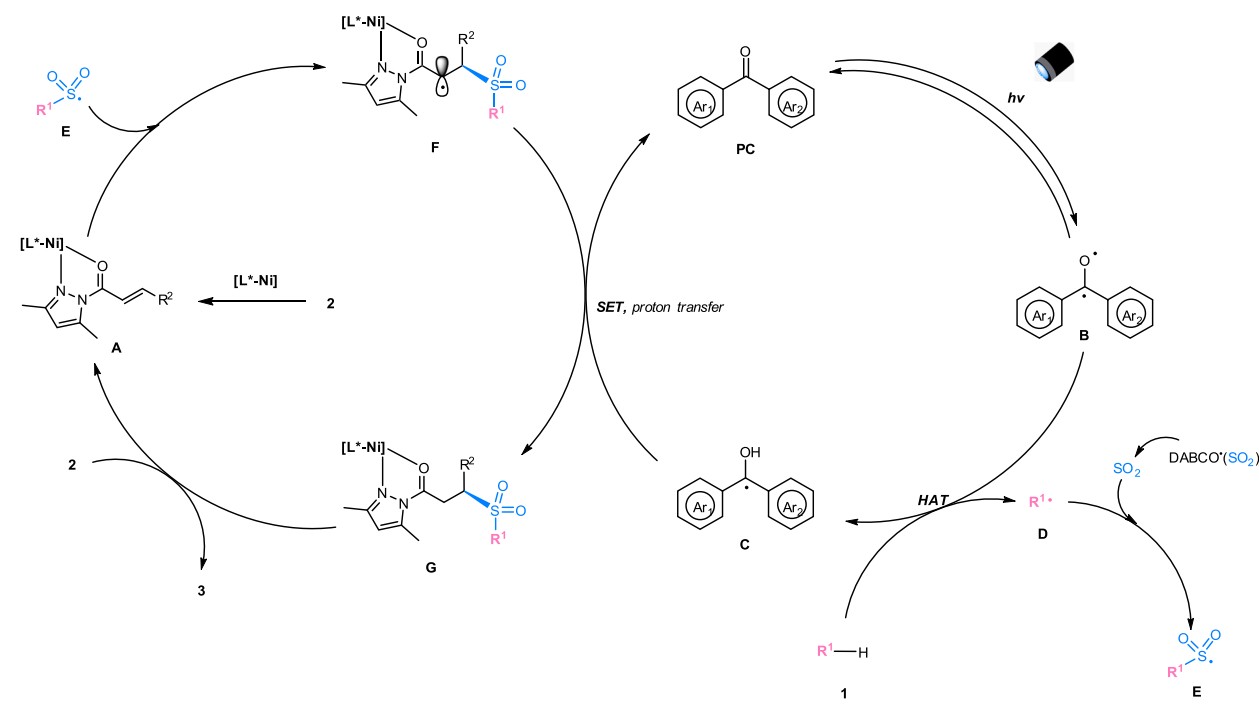

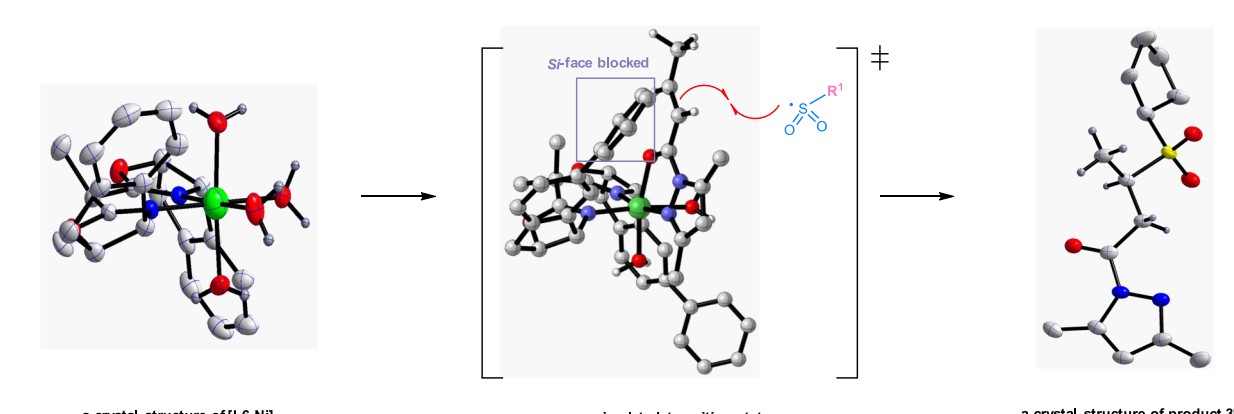

a crystal structure of [L6-Ni]          a simulated transition state          a crystal structure of product 3b

**Fig. 5 Proposed reaction mechanism. a** Proposed catalytic cycles. **b** Left: a crystal structure of chiral nickel complex **[L6-Ni]** (CCDC no. 2054187). Middle: a transition state (**[L7-Ni-2a]** with sulfonyl radical), simulated by Gaussian 09, and drawn by CYLview 1.0 (Fig. 5b, middle)[81]. Right: a crystal structure of product **3b** (CCDC no. 2028396). PC, photocatalyst; DABCO • (SO₂)₂, 1,4-Diazabicyclo[2.2.2]octane-1,4-diium-1,4-disulfinate; SET, single electron transfer; HAT, hydrogen atom transfer.

stabilized sulfonyl radical (**E**)[41]. Owing to the electronic and steric effects, **E** reacts with the metal-coordinated Michael acceptor (**A**) through an outer-sphere rather than an inner-sphere pathway, affording the radical complex (**F**)[60,62–66]. Such an outer-sphere attack might be critical to avoid side reactions such as self-coupling or elimination and to achieve a high level of asymmetric induction in the photochemical reaction[67–80]. Subsequent single electron transfer and proton transfer among intermediates **C**, **F** and a small amount of water in the solution lead to the formation of the neutral complex (**G**) and the organophotocatalyst (**PC**). Ultimately, ligand exchange between intermediate **G** and the substrate (**2**) gives the chiral sulfone product (**3**) and regenerates the coordinated α,β-unsaturated *N*-acylpyrazole (**A**). Besides, a pathway involving DABCO-participated reductive quench cannot be excluded (see more details in the Supplementary Fig. 12)[58].

A crystal structure of the nickel complex **[L6-Ni]** exhibits an octahedral geometry, in which the six coordination sites are

occupied by one chiral ligand and four water molecules (Fig. 5b, left). Accordingly, an intermediate **[L7-Ni-2a]** is simulated by Gaussian 09, and a proposed transition state is modeled by CYLview 1.0 (Fig. 5b, middle)[81]. The sulfonyl radical (**E**) interacts with the C=C double bond of the coordinated α,β-unsaturated *N*-acylpyrazole from *Re*-face with less steric hindrance, which is consistent with an observed *R*-configuration in product **3b** (Fig. 5b, right). The modeled transition state structure also illustrated that the extended phenyl substituents on the chiral ligand (**L7**) were critical for a high level of asymmetric induction.

**Synthetic utility**. A mmol-scale reaction was performed to demonstrate the synthetic utility of the reaction. A mixture of toluene (**1v**, 533 μL, 5.0 mmol), **2a** (164 mg, 1.0 mmol) and DABCO · (SO₂)₂ (180 mg, 0.75 mmol) was irradiated with a blue LEDs lamp under the standard conditions (Fig. 6a), leading to the production of 227 mg of **3v** (71% yield, 91% ee). The yield and

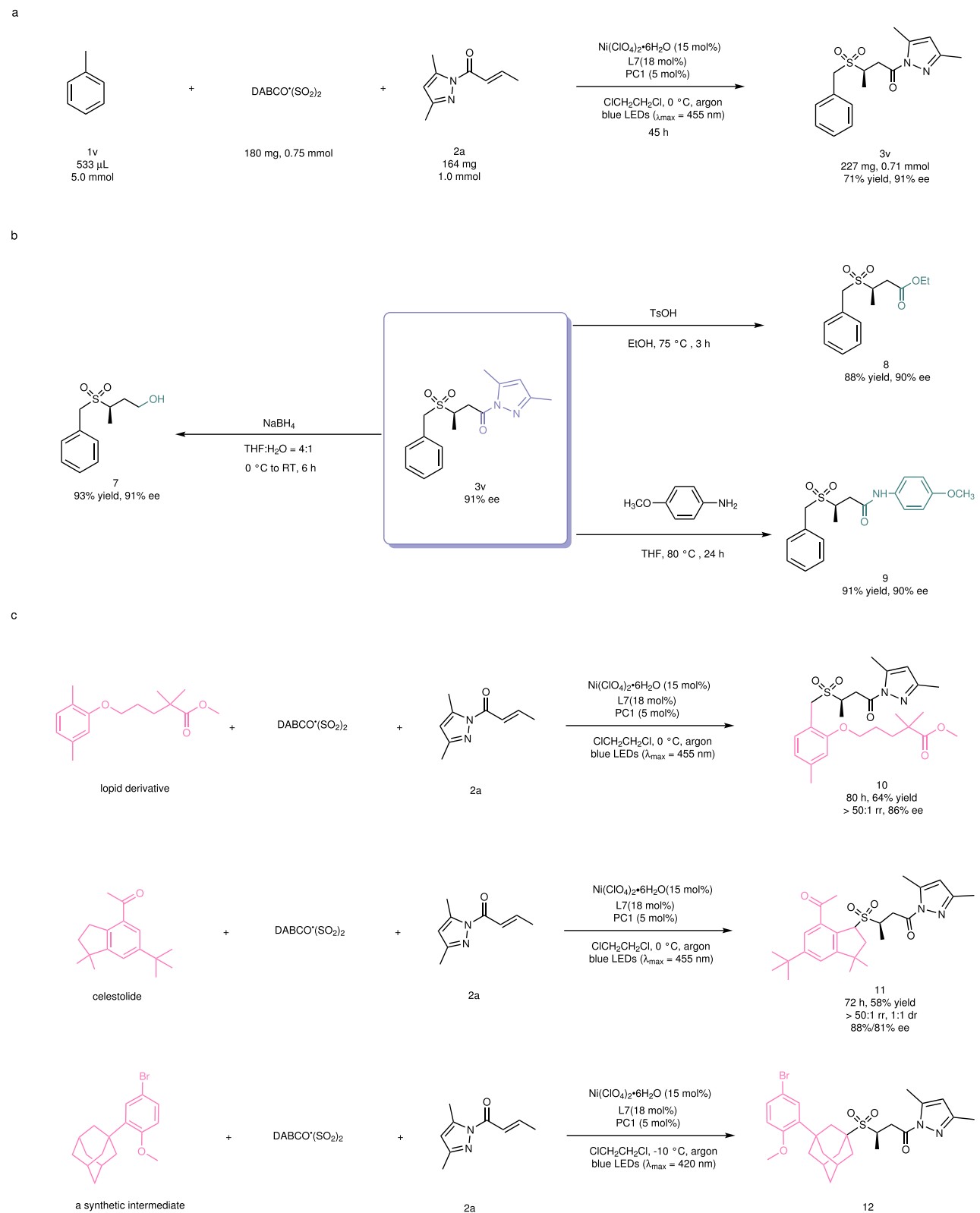

**Fig. 6 Synthetic utility of this method. a** A mmol-scale photocatalytic reaction. **b** Transformations of chiral sulfone product **3v**. **c** Late-stage modification of bioactive molecules. PC, photocatalyst; DABCO • (SO₂)₂, 1,4-Diazabicyclo[2.2.2]octane-1,4-diium-1,4-disulfinate; THF, tetrahydrofuran; TsOH, 4-Toluenesulfonic acid; RT, room temperature.

enantioselectivity were basically the same as in the small-scale reaction. Next, we investigated further transformations of the reaction product. For example, an alcohol derivative (**7**) was obtained in 93% yield and with 91% ee by treatment of the chiral sulfone (**3v**, 91% ee) with $NaBH_4$ in a mixed solvent of tetrahydrofuran (THF) and $H_2O$ at 0 °C to room temperature. **3v** could also be converted into the corresponding ester (**8**) or amide (**9**) in good yield and with retention of enantiomeric excess by substitution of the pyrazole moiety with an ethoxy or an amino group, respectively (Fig. 6b). Finally, the photochemical reaction was applied to the late-stage modification of bioactive molecules (Fig. 6c). Under the standard conditions, the reaction of a lopid derivative afforded its chiral sulfone derivative (**10**) as a single regioisomer in 64% yield and with 86% ee. The product was identified by $^1$H-NOESY NMR, suggesting that the *orth*-methyl group was selectively functionalized. We assumed that the excellent site-selectivity towards the $C(sp^3)$-H at the *orth*-position might be attributed to neighboring electronic effects of the heteroatom (oxygen) and its coordination ability to the nickel catalyst. Such precise recognition of two very similar benzylic C $(sp^3)$-H bonds further confirmed the powerful catalyst-control of selectivity in the photochemical reaction. Using a similar protocol, celestolide and a synthetic intermediate of differin could be converted to the corresponding sulfone products **11** and **12** with excellent regioselectivity and high enantioselecitivity, respectively.

## Discussion

In summary, we have developed an effective protocol for asymmetric sulfonylation reactions through direct $C(sp^3)$-H functionalization of cycloalkanes, alkanes, toluene derivatives, or ethers. The three-component reaction among an inert $C(sp^3)$-H precursor, an $SO_2$ surrogate (DABCO·$(SO_2)_2$), and a common Michael acceptor (α,β-unsaturated carbonyl compound) is enabled by dual organophotocatalysis and nickel-based asymmetric catalysis. A number of biologically interesting enantioenriched α-C sulfones are obtained with high selectivity under mild conditions (> 50 examples, up to 50:1 rr and 95% ee). The $SO_2$ surrogate provides radical stabilization access for suppression of the side reactions and perhaps being beneficial for stereocontrol in the photochemical reaction. We believe that it would provide an appealing opportunity to develop valuable asymmetric transformations starting from the abundant hydrocarbon compounds.

## Methods

**Preparation of a solution of the non-racemic nickel catalyst [L7-Ni] in dichloroethane.** A solution of Ni(ClO$_4$)$_2$·6H$_2$O (11.0 mg, 0.030 mmol) and non-racemic ligand **L7** (18.4 mg, 0.036 mmol) in 1,2-dimethoxyethane (DME, 4.0 mL) was stirred at 75 °C for 5 h, then the resulting solution was concentrated under reduced pressure to remove the solvent. The residue was redissolved in dichloroethane (DCE, 4.0 mL), which was used freshly as the metal catalyst for the photochemical reactions.

**Representative procedure for the photocatalytic asymmetric three-component reaction.** A dried 25 mL Schlenk tube was charged with **1a–1zo** (2.0 or 4.0 mmol), **2a** (0.20 mmol), **PC1** (3.4 mg, 0.010 mmol), DABCO·$(SO_2)_2$ (36.0 mg, 0.15 mmol), chiral nickel catalyst [**L7-Ni**] (4.0 mL, taken from the abovementioned freshly prepared solution in DCE), and DCE (4.0 mL). The mixture was degassed via three freeze-pump-thaw cycles. The Schlenk tube was positioned approximately 5 cm away from a 24 W blue LEDs lamp ($\lambda_{max}$ = 455 or 420 nm). After being stirred at 0 °C or 20 °C for 24–75 h (monitored by TLC analysis), the reaction mixture was concentrated, then purified by flash chromatography on silica gel (eluted with CH$_2$Cl$_2$ or PE:EtOAc = 4:1) to afford a non-racemic product **3a–3zo**, **5–12**.

## Data availability

The authors declare that the data supporting the findings of this study are available within this article and its Supplementary Information file, or from the corresponding author upon reasonable request. The experimental procedures and characterization of all

new compounds are provided in Supplementary Information. The X-ray crystallographic coordinates for structures reported in this study have been deposited at the Cambridge Crystallographic Data Centre (CCDC), under deposition numbers CCDC 2028396 (**3b**), and CCDC 2054187 ([**L6-Ni**]). These data can be obtained free of charge from The Cambridge Crystallographic Data Centre via www.ccdc.cam.ac.uk/data_request/cif.

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

## Acknowledgements

We gratefully acknowledge funding from the National Natural Science Foundation of China (grant no. 22071209), the Natural Science Foundation of Fujian Province of China (grant no. 2017J06006), and the Fundamental Research Funds for the Central Universities (grant no. 20720190048).

## Author contributions

S.C. and L.G. designed and conceived the project. S.C. and W.H. conducted all the synthetic and mechanistic experiments. S.C., W.H., and L.G. analyzed and interpreted the experimental data. Z.Y. designed and performed the theoretical calculations. L.G. prepared the manuscript. S.C. and Z.Y. prepared the Supplementary Information.

## Competing interests

The authors declare no competing interests.
