## [Peer Review File · Nature Communications]

REVIEWER COMMENTS

Reviewer #1 (Remarks to the Author):

The main innovation of this paper is in the development of a SO₂ source that can serve as a trap to control the reactivity of alkyl radicals, so that they can selectively add to alpha,beta-unsaturated amides. The finding that sulfonyl radical additions to Michael-type acceptors can be carried out with high enantioselectivity using 15% loading of a Lewis acid catalyst is a remarkable finding that deserves publication. However, the requirement for SO₂ is somewhat limiting, because sulfone functionality is rarely part of the final target in most synthetic chemistry efforts. Although sulfones are fairly common intermediates for synthesis, in such cases the stereogenic center at the alpha-carbon is generally deprotonated (e.g., for Julia olefination), which would destroy the stereogenic center created in this chemistry. Therefore it is questionable whether the innovative aspects of this chemistry may be limited to specialized cases, and therefore not as widely impactful as most of the synthetic chemistry that is published in such a high-impact venue. With this in mind, I recommend publication in a more specialized high-impact chemistry journal, such as *J. Am. Chem. Soc.* or *Angew. Chem. Int. Ed.*

The DABCO is not necessarily an innocent bystander in such reactions. It could also be proposed to serve as a reductant for excited photocatalyst B in Figure 5, with the resulting DABCO radical cation conducting the H-atom abstraction. This role for DABCO has been used previously (for example, see ref 58).

Fig 6: Selectivity in preparation of compound 10 is remarkable, and deserves further explanation or comment. It is also important for the authors to explain how they have identified which methyl group has been functionalized. The other regioisomer would presumably have very similar spectral characteristics.

The authors state that 3b was determined to be R configuration, and that all other products were assigned R by analogy. However some products from radical acceptors with branched or aryl groups at the beta carbon have changed the priorities of the substituents in comparison with 3b. Thus, the same mechanism may produce S configuration, simply by virtue of changing the nomenclature priorities of the groups. For each compound the authors should clarify this in the experimental writeup by adding a statement such as "The configuration was assigned S by analogy to (R)-3b."

Figure 4a: give yield of TEMPO adduct

Last sentence of p12: duplicated text "quenching experiments"

Reviewer #2 (Remarks to the Author):

This paper describes a catalytic three component coupling of alkanes, an SO₂ surrogate, and Michael acceptors by combination of a photoredox catalyst and a chiral nickel complex catalyst. The authors previously reported that PC1 is a good photoredox catalyst for alkane C-H abstraction. However, the generated alkyl radical was not reactive to the Michael acceptors. Therefore, authors devised to trap

the alkyl radical by SO₂ to generate alkyl sulfonyl radical, which was reactive enough for Michael addition under chiral Lewis acid catalysis. The products are useful for the synthesis of biologically active compounds.

This is a clever way to apply alkyl radicals existing in low concentration generated from stable C-H bonds, to a synthetically useful reaction. Therefore, this paper can be accepted after revising the manuscript based on the following comments.

1. DABCO(SO₂)₂ was specially reactive in this method. Can the authors comment on possible reasons?
2. Decatungstate PC4 is a well-known photoredox catalyst for alkane hydrogen atom abstraction. Yet, no desired product was obtained. Can the authors explain possible reasons?
3. In the paragraph before Fig.3, the authors attributed the moderate ee obtained for beta-aryl substituted substrates to Z/E isomerization. Authors should subject the isolated Z isomer to the reaction conditions and confirm if the hypothesis is right.
4. For the simulated transition state shown in Fig 5b, did the authors calculate energies of other isomers? How feasible is another isomer where the nitrogen atom of the pyrazole coordinates to the axial site of the nickel and the carbonyl oxygen atom coordinates to the planar site, which should afford the enantiomer, or both oxygen and nitrogen atoms coordinate to the planar sites, which should afford racemic products?

Reviewer #3 (Remarks to the Author):

In the manuscript, a nice protocol for a three component coupling involving C-H bonds, sulfur dioxide and alkenes is described. A key method of the C-H activation using a fused tetraketone is taken from a previous paper by the same group (ref 38, *Nat. Catal.* 2, 1016–1026 (2019)). Therefore, the novelty is that authors applied a known radical generation method to another known process of trapping sulfur dioxide. Asymmetric sulfur dioxide radical reactions have also been known (for example, ref *Adv. Synth. Catal.* 360, 1060). Nevertheless, the manuscript is very well done. Good yields and high enantioselectivities are attained, and the paper is in a fashionable area of photocatalysis. The obtained experimental data support the proposed mechanism, and in terms of general presentation, the work is methodologically sound. Therefore, despite the lack of conceptual ideas, the paper would likely be well accepted by synthetic community. Overall, the manuscript can be accepted.

Point-by-Point Response to the Reviewers' Comments

Responses to the comments of reviewer 1.

(1) The DABCO is not necessarily an innocent bystander in such reactions. It could also be proposed to serve as a reductant for excited photocatalyst **B** in Figure 5, with the resulting DABCO radical cation conducting the H-atom abstraction. This role for DABCO has been used previously (for example, see ref 58).

Our response:

We do appreciate this highly valuable suggestion. Indeed, it is possible that DABCO can serve as a reductant to quench the excited state of the photocatalyst (**B**, Figure 5). In the revised Supplementary Information, we add an alternative catalytic pathway to section 5.8 (Supplementary Fig. 12, page S97). This is also mentioned in the main text as ‘Besides, a pathway involving DABCO-participated reductive quench cannot be excluded (see more details in the Supplementary Fig. 12).⁵⁸’ (page 14 of the manuscript). We also test the reaction of $1\mathbf{a} + \text{DABCO}(\text{SO}_2)_2 + 2\mathbf{a} \rightarrow 3\mathbf{a}$ in the presence of an iridium photocatalyst (the similar catalytic system as reference 58), and it failed to afford any desired product, suggesting that the pathway involving DABCO-participated reductive quench might not be the main mechanism. This result is also added in the Supplementary Information (page S98).

A pathway involving DABCO-participated reductive quench.

A control reaction of 1a+ DABCO(SO₂) + 2a → 3a with the catalyst system in ref. 58.

(2) Fig 6: Selectivity in preparation of compound **10** is remarkable, and deserves further explanation or comment. It is also important for the authors to explain how they have identified which methyl group has been functionalized. The other regioisomer would presumably have very similar spectral characteristics.

Our response:

Thanks for the comment. In the revised manuscript, we add the statement 'The product was identified by ¹H-NOESY NMR, suggesting that the *ortho*-methyl group was selectively functionalized. We assumed that the excellent site-selectivity towards the C(sp³)-H at the *ortho*-position might be attributed to neighboring electronic effects of the heteroatom (oxygen) and its coordination ability to the nickel catalyst.' in the main text (page 16).

¹H-NOESY NMR of product **10**.

(3) The authors state that **3b** was determined to be *R* configuration, and that all other products were assigned *R* by analogy. However some products from radical acceptors with branched or aryl groups at the beta carbon have changed the priorities of the substituents in comparison with **3b**. Thus, the same mechanism may produce *S* configuration, simply by virtue of changing the nomenclature priorities of the groups. For each compound the authors should clarify this in the experimental writeup by adding a statement such as "The configuration was assigned *S* by analogy to (*R*)-**3b**."

Our response:

We sincerely thank this important point. Indeed, product **3zs** and **3zu–3zy** should be assigned as *S* configuration according to the Cahn-Ingold-Prelog rule. In the revised Supplementary Information, we added the statement 'The configuration of xx was assigned xx by analogy to (*R*)-**3b**.' for each product in the experimental section.

(4) Figure 4a: give yield of TEMPO adduct.

Our response:

In the revised manuscript, we add the yield of TEMPO adduct (trace) in Figure 4a.

(5) Last sentence of p12: duplicated text "quenching experiments".

Our response:

We apologize for our carelessness. It has been corrected in the revised manuscript. Moreover, we have carefully checked all the Figures and Schemes to avoid this type of mistakes.

Responses to the comments of reviewer 2.

(1)The products are useful for the synthesis of biologically active compounds.

This is a clever way to apply alkyl radicals existing in low concentration generated from stable C-H bonds, to a synthetically useful reaction. Therefore, this paper can be accepted after revising the manuscript based on the following comments.

Our response:

We sincerely thank the positive evaluation on our manuscript by reviewer 2. The highly useful comments and

suggestions encourage us to look into the reaction system in more depth.

(2) DABCO(SO₂)₂ was specially reactive in this method. Can the authors comment on possible reasons?

Our response:

We believe that two possible reasons can explain the special reactivity in this reaction. One is that in comparison to other SO₂ source such as Na₂S₂O₅, DABCO(SO₂)₂ has better solubility in ClCH₂CH₂Cl which can release SO₂ at an appropriate concentration to trap the alkyl radicals. In addition, a pathway involving DABCO-participated reductive quench cannot be completely excluded. We add the alternative reaction mechanism in the Supplementary Information (page S97).

A pathway involving DABCO-participated reductive quench.

(3) Decatungstate **PC4** is a well-known photoredox catalyst for alkane hydrogen atom abstraction. Yet, no desired product was obtained. Can the authors explain possible reasons?

Our response:

Decatungstate **PC4** is indeed a very useful catalyst for alkane hydrogen atom abstraction. However, it cannot be dissolved in our reaction solvent (ClCH₂CH₂Cl). In other solvents of better solubility, such as methanol and acetonitrile, the reaction with **PC4** as the photocatalyst did not proceed as well. This might be due to the different coordination behaviors and catalytic activity of the nickel catalysts in the solvents with coordinating capability.

(4) In the paragraph before Fig.3, the authors attributed the moderate ee obtained for beta-aryl substituted substrates to

Z/E isomerization. Authors should subject the isolated Z isomer to the reaction conditions and confirm if the hypothesis is right.

Our response:

We examined the isolated Z-isomer to the photocatalytic reaction under the standard conditions. Consequently, significantly reduced yield (52%) and ee value (44%) were obtained, which is consistent with the hypothesis. We add this result in the Supplementary Information (page S85).

(5) For the simulated transition state shown in Fig 5b, did the authors calculate energies of other isomers? How feasible is another isomer where the nitrogen atom of the pyrazole coordinates to the axial site of the nickel and the carbonyl oxygen atom coordinates to the planar site, which should afford the enantiomer, or both oxygen and nitrogen atoms coordinate to the planar sites, which should afford racemic products?

Our response:

We do appreciate this important comment which encourages us to look into the reaction system in more depth. Other possible isomers were simulated as below. The geometry optimization of these isomers was added in Supplementary Information (pages S88–S97). However, the energy difference among the isomers strongly depends on the other coordinated atoms or ligands. Once one chiral ligand and one substrate coordinate with the nickel, there are still two free coordination sites, which can be occupied with many possibilities. The radical process makes it a more challenging problem to explain the selectivity from an energy perspective. Thereby, we only use the transition state simulated on the basis of crystal structure of the catalyst and product to account for it. In the future, we will cooperate with the experts in the field of crystallography and theoretical chemistry to understand the system better.

Different coordination models

Responses to the comments of reviewer 3.

In the manuscript, a nice protocol for a three component coupling involving C-H bonds, sulfur dioxide and alkenes is described. A key method of the C-H activation using a fused tetraketone is taken from a previous paper by the same group (ref 38, Nat. Catal. 2, 1016–1026 (2019)). Therefore, the novelty is that authors applied a known radical generation method to another known process of trapping sulfur dioxide. Asymmetric sulfur dioxide radical reactions have also been known (for example, ref Adv. Synth. Catal. 360, 1060). Nevertheless, the manuscript is very well done. Good yields and high enantioselectivities are attained, and the paper is in a fashionable area of photocatalysis. The obtained experimental data support the proposed mechanism, and in terms of general presentation, the work is methodologically sound. Therefore, despite the lack of conceptual ideas, the paper would likely be well accepted by synthetic community. Overall, the manuscript can be accepted.

Our response:

We sincerely thank the highly positive evaluation on our manuscript and encouragement from reviewer 3. Moreover, we have carefully checked the manuscript and Supplementary Information to avoid typos and improve their quality.

Finally, we would like to express our sincere appreciation to the referees and editors for your kind help and support during the manuscript submission and review process. If you need any further information, please do not hesitate to contact me at any time.

REVIEWERS' COMMENTS

Reviewer #1 (Remarks to the Author):

The authors have carefully and thoroughly addressed the reviewers' comments, and the manuscript is now improved. I recommend acceptance.

Reviewer #2 (Remarks to the Author):

According to the authors' revision, all my initial concerns are clear now. This paper can be accepted.